# PCR enables rapid detection of dermatophytes in practice

E. Aho-Laukkanen,[1] V. Mäki-Koivisto,[1] J. Torvikoski,[2] S. P. Sinikumpu,[3] L. Huilaja,[3] I. S. Junttila[1,4,5,6]

**ABSTRACT** Dermatophytes cause superficial infections of skin, hair, and nails. Even though they rarely cause severe infections, they are relatively common, particularly in primary health care. Diagnosis of dermatophyte infections has relied on relatively slow culture-based methods. Nucleic acid-based detection (PCR) methods might provide results more rapidly. Here, we describe the transition from culture-based methods into PCR-based methods in Northern Finland with a catchment area of approximately 720,000 mostly Caucasian people. This study included 14,330 samples collected between 2019 and 2022. Our results showed that the PCR-based method has become the diagnostic test of choice for these infections in this area. Commercial real-time PCR assay DermaGenius 2.0 complete multiplex test detected more positive results than culture and covered the most important dermatophytes, *Candida albicans*, and few less common species. With PCR, the mean turn-around-time from sample request to results decreased from 19 days to 16 hours.

**IMPORTANCE** Superficial fungal infections, dermatophytosis, are remarkably common worldwide, affecting an estimated 20%–25% of the global population. In the diagnosis of these infections, fast and accurate results by PCR shorten the time to diagnosis and help clinicians to avoid unnecessary antifungal treatments.

**KEYWORDS** dermatophytes, mycology, nucleic acid technology, PCR, culture

Dermatophytes are molds that cause infections in keratinized layers of the body, such as the skin and nail. Dermatomycoses are remarkably common, affecting an estimated 20%–25% of the population worldwide (1, 2). In Finnish primary health care, fungal infections are among the 10 most common dermatological indications for physician consultation (3). In a Finnish cohort study among middle-aged adults, 26.8% had tinea pedis, and 9.4% had onychomycosis (4). Among people aged >70 years, these infections are even more common with prevalences of 48.6% and 29.9%, respectively (5).

Three genera of dermatophytes are the major causes of tinea infections: *Trichophyton, Microsporum,* and *Epidermophyton* (6). Other genera are *Arthroderma, Ctenomyces, Lophophyton*, and *Nannizzia* (7). In addition, two new genera have been introduced: *Guarromyces* and *Paraphyton* (7). In Finland, *Trichophyton rubrum* and *Trichophyton mentagrophytes* complex (including *Trichophyton interdigitale*) are the most common causes for tinea pedis, tinea cruris, tinea corporis, and onychomycosis, whereas *Trichophyton violaceum* and *Microsporum audouinii* are the predominant dermatophytes involved in tinea capitis (8).

The diagnosis of a dermatophyte infection should always be confirmed with a clinical sample sent for microbiological analysis (8). Traditionally, superficial fungal infection has been verified with microscopic examination and fungal culture. As dermatophytes grow slowly, it often takes 1 to 6 weeks to complete the results with these methods. Compared to fungal culture, nucleic acid-based methods are quicker and more sensitive (9–11). In addition, nucleic acid amplification techniques allow analysis of several samples simultaneously.

Address correspondence to E. Aho-Laukkanen, elina.aho-laukkanen@nordlab.fi.

The authors declare no conflict of interest.

See the funding table on p. 8.

To shorten the turn-around time (TAT), an in-house polymerase chain reaction (PCR) test (9), including specific primers for *T. rubrum*, *T. mentagrophytes*, and general pan-dermatophyte target, was introduced at Northern Finland Laboratory Centre (NordLab) in 2015. In 2018, a commercial real-time PCR assay DermaGenius (DG) 2.0. complete multiplex test (PathoNostics, Amsterdam, The Netherlands) became commercially available as a CE-IVD certified kit. Compared with the in-house PCR, DG has a wider panel of species, detecting several dermatophytes as well as *Candida albicans*. The in-house PCR test of NordLab was replaced with DG in May 2019 to cover more dermatophytes and to decrease the demand for culture and microscopic examination. We evaluate here the quantitative and qualitative aspects of this transformation process from culture-based diagnostics to PCR-based method in the detection of superficial fungal infections.

## MATERIALS AND METHODS

This study was a retrospective observational analysis under a permit number 865 of North Ostrobothnia wellbeing services county, Pohde (Finland).

### Samples

We analyzed the results of 14,330 routine skin, nail, and hair samples collected from patients with clinical suspicion of superficial fungal infection in Northern Finland (NordLab laboratory area covering approximately 720,000 inhabitants) between 2019 and 2022. The samples were analyzed with a commercial PCR test (DG) or with culture and microscopy in clinical microbiology laboratories according to clinician's decision.

### DNA extraction and nucleic acid amplification

Samples were pre-treated with 180 µL of Buffer ASL (Qiagen, Hilden, Germany) in 2 mL Lyse&Spin basket (Qiagen, Hilden, Germany) and incubated overnight at room temperature. The next day, 20 µL of Proteinase K solution (Qiagen, Hilden, Germany) was added, and the samples were incubated 30 min at 65°C. After a brief centrifugation, 400 µL of distilled water was added, and the samples were centrifuged at 13,300 rpm for 10–15 min. DNA extraction was performed with QIAsymphony DSP Virus/Pathogen Mini Kit (Qiagen, Hilden, Germany) and SP instrument (Qiagen, Hilden, Germany). The extraction method was verified with the manufacturer.

PCR with DermaGenius (DG) 2.0. complete multiplex kit (PathoNostics, Amsterdam, The Netherlands) was run on the Rotor-Gene Q (Qiagen, Hilden, Germany) according to the manufacturer's instructions. The kit contained the PCR primers and probes for *C. albicans, T. interdigitale, T. mentagrophytes, T. tonsurans, T. violaceum* and *T. rubrum/T. soudanense*, *T. benhamiae, T. verrucosum, M. canis, M. audouinii*, and *E. floccosum*. The internal control (IC) was used to monitor the extraction and amplification processes, PCR inhibition, and false-negative results. PCR results were analyzed with melting curve analysis on the Rotor-Gene software (version 2.3.5).

The performance of PCR was verified by using fungal ATCC strains, fungal quality control strains (UK NEQAS), and patient samples that were initially examined with culture and microscopy. The DermaGenius PCR test is CE-IVD certified by the manufacturer.

### Culture and microscopic examination

Cultures were performed in accredited clinical mycology laboratories: in NordLab (Oulu, Finland) until May 2019, and in HUS Diagnostic Center (Helsinki, Finland) since May 2019. For all non-swab samples, the laboratories performed microscopic examination with potassium hydroxide digestion followed by Calcofluor White staining (Becton Dickinson, Franklin Lakes, New Jersey, US) to detect fungal structures directly from the sample. Calcofluor White staining was omitted if the sample was too scant for both culture and microscopic examination.

In NordLab clinical mycology laboratory, all samples were cultured on Sabouraud Dextrose Agar (SDA) plates with chloramphenicol (Thermo Fisher Scientific, Waltham, Massachusetts, US), and dermatophyte test medium (DTM) plate with chloramphenicol and cycloheximide. Additionally, skin and hair samples were cultured on Dixon Agar plate. Agar plates were incubated 30–35°C for up to 6 weeks. In HUS clinical mycology laboratory, all samples were cultured on Sabouraud Maltose Agar (SMA) plates, consisting of Sabouraud Maltose Agar (Biolife, Milan, Italy) with chloramphenicol and gentamycin additives, and SDA plates, consisting of Sabouraud Dextrose agar (Thermo Fisher Scientific, Waltham, Massachusetts, US) with chloramphenicol, gentamycin, and cycloheximide additives. SMA and SDA plates were incubated at 28°C for 5 weeks total.

Fungal identification was based on the macromorphological characteristics of fungal colonies and their micromorphological characteristics in Lactophenol Cotton Blue wet-mount preparations. Matrix-assisted laser desorption ionization time-of-flight (MALDI-TOF) mass spectrometry (VITEK MS, bioMérieux, Marcy-l'Étoile, France) was used routinely in addition to morphological identification. If necessary, the identification was confirmed with PCR (ITS1 and ITS2 regions) and fungal genome sequencing in HUS Diagnostic Center, Finland (12).

## Statistical analysis

The trends in conducted test detecting the superficial fungal infection using nucleic acid detection and traditional methods were analyzed using the Mann-Kendall trend test. Statistical difference of positivity rates was analyzed with Chi-squared test. A $P$ value of < 0.001 was considered statistically significant.

## RESULTS

The total number of samples was between 4,000 and 5,000 per year during 2015–2022, except during COVID-19 pandemic years 2020–2021 (Fig. 1). From 2015 to 2022, the amount of PCR samples grew steadily, and from 2018 to 2022, the amount of culture and microscopic examination samples decreased (Fig. 1). Both the increase in PCR samples and the decrease in culture and microscopic examination samples were statistically significant with Mann-Kendall trend test. The number of PCR samples exceeded culture and microscopic examination samples in 2019 (Fig. 1). In 2022, 87.6% of the superficial fungal infection samples were examined with PCR.

*T. rubrum* (78.8%) and *T. interdigitale* (7.3%) were the most common species in superficial fungal infection samples during the study period with both PCR and culture methods (Table 1). The positivity rates of PCR were significantly higher ($P$ < 0.001) than the rates of culture methods in every year during 2019–2022 (Fig. 2). With PCR, the average TAT was 16 h (0.7 days), whereas with culture it was 19 days in 2019–2022 (Table 2).

DG detected 94.3% (5743) of all positive results (6088) found with PCR and culture methods (Rows 1–11, Table 1). Some results were found only with culture and/or microscopic examination (Rows 12–22, Table 1). These results were mainly 1) fungal mycelium in microscopic examination but no fungal growth in culture (2.5% of all positive results), and 2) other *Candida* species than *C. albicans* (1.3% of all positive results). The other *Candida* species than *C. albicans* was most commonly *C. parapsilosis* (data not shown).

## DISCUSSION

Since the introduction of in-house dermatophyte PCR test in 2015, the number of PCR samples has been in constant growth, whereas the number of culture and microscopic examination samples has decreased. After introducing the commercial DG PCR test in 2019, the number of PCR samples exceeded culture and microscopic examination samples as nucleic acid detection became the preferred method for detecting dermatophytes from superficial fungal infection samples. The performance of DG as compared

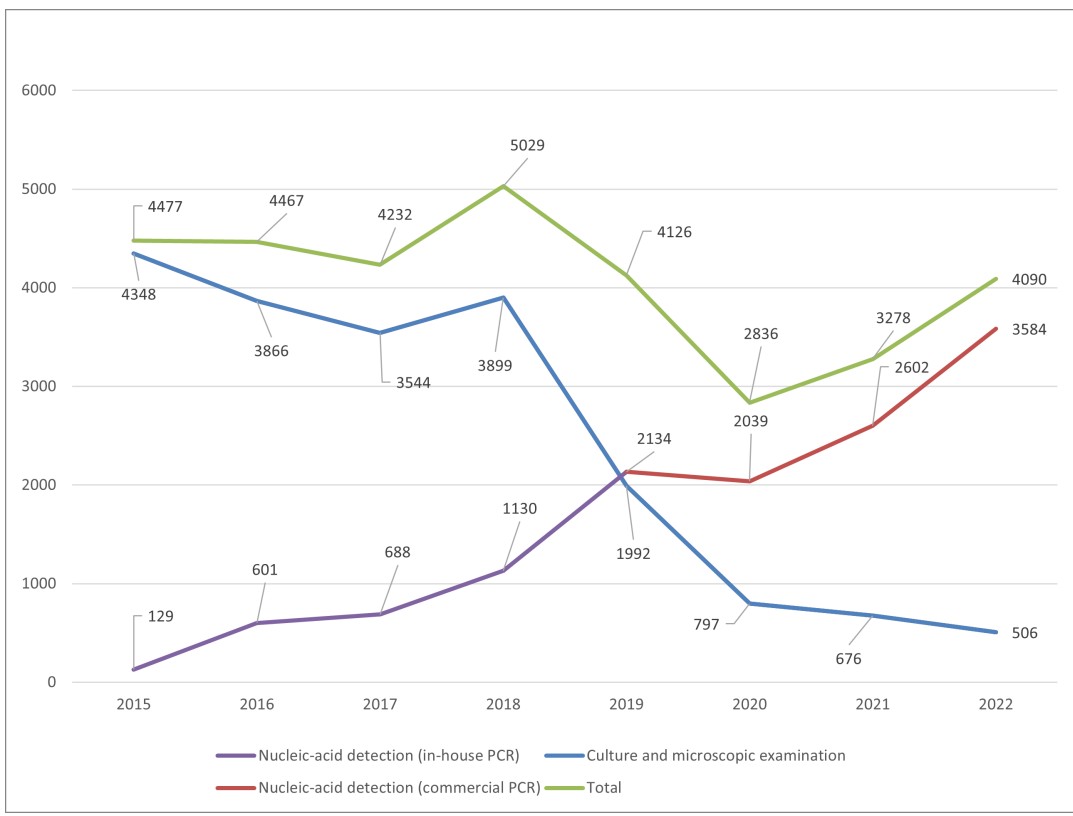

**FIG 1** Numbers of superficial fungal infection samples in 2015–2022 (NordLab, Finland) and their detection methods.

with culture has been previously described by Trovato and colleagues (13, 14) and was also verified in our laboratory (Table S1). In our verification, the results of ATCC strains (*n* = 4), UK NEQAS quality control strains (*n* = 7), and four of five patient samples were identical with culture and DG (Table S1).

The number of positive results, especially for *T. rubrum* and *T. interdigitale,* increased as the use of PCR test became more common (Fig. 1; Table 1). Most likely this increase reflects the high sensitivity of the used method: a study conducted in Senegal and in Belgium (15) found out that DermaGenius 2.0 was significantly more sensitive than culture (*P* < 0.001). However, they analyzed only hair samples in their study. As *T. rubrum* and *T. interdigitale* are the most common causes of tinea pedis and onychomycosis (16, 17), the use of PCR method may help to make more accurate diagnoses.

In this study, we found that with PCR, the mean TAT from sample request to results dropped from 19 days to 16 h. Fast and accurate results of PCR shorten the time to diagnosis and help clinicians to avoid possibly unnecessary empirical antifungal treatments that may have side effects (18). The National Finnish guideline recommends topical treatment of tinea infections between the toes without diagnostic sampling (8). However, for all other instances, PCR or culture sample needs to be taken to identify the causative agent (8). Another significant benefit of PCR compared to culture is that PCR samples can be analyzed while a patient has already received antifungal treatment. Instead, in culture, a fungus must be alive in a sample, and thus a break from 2 weeks up to 6 months from antifungal treatment to sampling is required (8). However, as PCR can detect DNA also from non-viable fungal cells, it is not suitable for following the efficacy of antifungal treatment. This has been mentioned in the national Finnish guideline as well (8).

The clinical usefulness of the DG assay in diagnostics of fungal infection has also been emphasized in other studies (14, 15, 19). Hayette et al. (19) pointed out that the advantages of DG included the detection of dermatophyte DNA even in case of

**TABLE 1** The positive results of superficial fungal infection samples with nucleic acid detection (PCR) and culture methods in 2019–2022 and their proportion (%) of all samples[a]

| | Result | PCR | % (all samples) | Traditional methods | % (all samples) | PCR | % (all samples) | Traditional methods | % (all samples) | PCR | % (all samples) | Traditional methods | % (all samples) | PCR | % (all samples) | Traditional methods | % (all samples) | Summary |
|---|---|---|---|---|---|---|---|---|---|---|---|---|---|---|---|---|---|---|
| 1 | *Trichophyton rubrum* | 654 | 15.9 | 335 | 8.1 | 820 | 28.9 | 165 | 5.8 | 1157 | 35.3 | 171 | 5.2 | 1409 | 34.4 | 89 | 2.2 | 4800 |
| 2 | *Trichophyton interdigitale* | 82 | 2.0 | 3 | 0.1 | 78 | 2.8 | 13 | 0.5 | 110 | 3.4 | 10 | 0.3 | 145 | 3.5 | 6 | 0.1 | 447 |
| 3 | *Candida albicans* | 37 | 0.9 | 45 | 1.1 | 38 | 1.3 | 27 | 1.0 | 65 | 2.0 | 32 | 1.0 | 71 | 1.7 | 29 | 0.7 | 344 |
| 4 | *Trichophyton violaceum* | 7 | 0.2 | 1 | 0.0 | 17 | 0.3 | 1 | 0.0 | 6 | 0.2 | 3 | 0.1 | 3 | 0.1 | 0 | 0 | 38 |
| 5 | *Epidermophyton floccosum* | 5 | 0.1 | 3 | 0.1 | 9 | 0.3 | 0 | 0 | 4 | 0.1 | 1 | 0 | 11 | 0.3 | 0 | 0 | 33 |
| 6 | *Trichophyton mentagrophytes* | 1 | 0.0 | 28 | 0.7 | 1 | 0.0 | 0 | 0 | 0 | 0 | 0 | 0 | 0 | 0 | 0 | 0 | 30 |
| 7 | *Trichophyton tonsurans* | 4 | 0.1 | 4 | 0.1 | 1 | 0.2 | 0 | 0 | 2 | 0.06 | 2 | 0.06 | 5 | 0.1 | 0 | 0 | 18 |
| 8 | *Trichophyton verrucosum* | 0 | 0 | 3 | 0.1 | 6 | 0.2 | 2 | 0.07 | 2 | 0.06 | 1 | 0.03 | 2 | 0.05 | 1 | 0.02 | 17 |
| 9 | *Trichophyton benhamiae* | 0 | 0 | 0 | 0 | 2 | 0.1 | 0 | 0 | 4 | 0.1 | 0 | 0 | 2 | 0.05 | 0 | 0 | 8 |
| 10 | *Microsporum canis/ ferrugineum* | 0 | 0 | 0 | 0 | 1 | 0.1 | 0 | 0 | 3 | 0.09 | 0 | 0 | 1 | 0.02 | 0 | 0 | 5 |
| 11 | *Microsporum audouinii* | 0 | 0 | 0 | 0 | 3 | 0.1 | 0 | 0 | 0 | 0 | 0 | 0 | 0 | 0 | 0 | 0 | 3 |
| 12 | Fungal mycelium in microscopic examination, culture negative | na | na | 14 | 0.3 | na | na | 75 | 2.6 | na | na | 60 | 1.8 | na | na | 1 | 0.02 | 150 |
| 13 | Other *Candida* sp | na | na | 49 | 1.2 | na | na | 12 | 0.4 | na | na | 8 | 0.2 | na | na | 11 | 0.3 | 80 |
| 14 | Other yeasts than *Candida* sp | na | na | 46 | 1.1 | na | na | 8 | 0.3 | na | na | 4 | 0.1 | na | na | 0 | 0 | 58 |
| 15 | Yeast cells in microscopic examination, culture negative | na | na | 12 | 0.3 | na | na | 8 | 0.3 | na | na | 7 | 0.2 | na | na | 5 | 0.1 | 32 |
| 16 | *Aspergillus* sp | na | na | 8 | 0.2 | na | na | 0 | 0 | na | na | 0 | 0 | na | na | 0 | 0 | 8 |
| 17 | *Scopulariopsis* sp | na | na | 3 | 0.1 | na | na | 1 | 0.04 | na | na | 0 | 0 | na | na | 1 | 0.02 | 5 |
| 18 | Other molds | na | na | 3 | 0.1 | na | na | 0 | 0 | na | na | 0 | 0 | na | na | 1 | 0.02 | 4 |
| 19 | *Fusarium* sp | na | na | 3 | 0.1 | na | na | 0 | 0 | na | na | 0 | 0 | na | na | 0 | 0 | 3 |

TABLE 1 The positive results of superficial fungal infection samples with nucleic acid detection (PCR) and culture methods in 2019–2022 and their proportion (%) of all samples[a] (Continued)

| Result | PCR | % (all samples) | Traditional methods | % (all samples) | PCR | % (all samples) | Traditional methods | % (all samples) | PCR | % (all samples) | Traditional methods | % (all samples) | PCR | % (all samples) | Traditional methods | % (all samples) | Summary |
|---|---|---|---|---|---|---|---|---|---|---|---|---|---|---|---|---|---|
| 20 Dermatophyte (without genus level id) | na | na | 0 | 0 | na | na | 1 | 0.04 | na | na | 1 | 0,03 | na | na | 0 | 0 | 2 |
| 21 Other *Microsporum* sp | na | na | 1 | 0.02 | na | na | 1 | 0.04 | na | na | 0 | 0 | na | na | 0 | 0 | 2 |
| 22 *Acremonium* sp | na | na | 0 | 0 | na | na | 1 | 0.04 | na | na | 0 | 0 | na | na | 0 | 0 | 1 |
| 23 Other *Trichophyton* sp | na | na | 0 | 0 | na | na | 1 | 0.04 | na | na | 0 | 0 | na | na | 0 | 0 | 1 |
| Summary | 790 | na | 561 | | 976 | na | 316 | | 1,353 | na | 300 | | 1,649 | na | 144 | | 6,089 |
| Total number of samples (positive and negative) | 2,134 | na | 1,992 | | 2,039 | na | 797 | | 2,602 | na | 676 | | 3,584 | na | 506 | | 14,330 |

[a]Traditional methods, culture and microscopic examination; Results 1–11 belong to used PCR test, results 12–22 are found only by Traditional methods; na, not applicable.

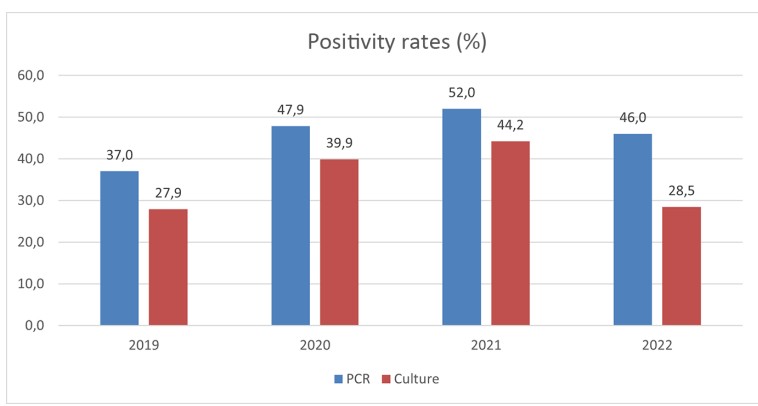

**FIG 2** The positivity rates of PCR and culture methods from 2019 to 2022.

contamination/colonization with non-dermatophyte molds, as well as a TAT around 1 day to obtain the results. In our study, the mean TAT was even lower (16 h). Also, Trovato et al. (14) found out that DG could reduce inadequate antifungal drugs and consequential side effects affecting patients' health or spreading of possible resistance. In Trovato et al. studies (13, 14), the sensitivity, specificity, positive predictive value, and negative predictive value of DG in the cutaneous dermatophytosis were 94.7%, 78.8%, 88.5%, and 89.6%, respectively. For diagnosis of onychomycosis the sensitivity, specificity, positive and negative predictive values of DG were 78.5%, 100%, 100%, and 75.9%, respectively.

Since the introduction of DG in 2019, only a few (<10 per species/4 years) other dermatophytes, yeasts, and molds were found with culture alone. Those species did not belong to the PCR test panel, and thus could not be found with PCR (Table 1). Therefore, based on the results of this study, DG covers the most important and frequent pathogens in Northern Finland. Also, according to the Finnish national guideline, a diagnosis of superficial fungal infection can be done with PCR without other confirmational methods, which is in line with our process (8). However, internationally, the role of PCR as a first step of dermatophyte diagnosis is still under discussion (13). It is also important to underline that if PCR test gives a negative result and clinical suspicion of fungal infection is strong, a culture sample must be taken to detect rare pathogens that the assay is not designed to detect, such as non-dermatophyte molds. Noteworthy, PathoNostics (Amsterdam, The Netherlands) has released a new version of DG (3.0), including *C. parapsilosis, Nannizzia gypsea, Scopulariopsis brevicaulis,* and a general pan-dermatophyte target. With the opportunity to detect these species, the PCR test could replace the culture method even more comprehensively as *C. parapsilosis* was the second most common *Candida* species after *C. albicans* in our study. Though C. *albicans* is also part of normal microbiota of skin, thus raising a concern of contamination, the number of *C. albicans* results did not increase strikingly with PCR compared to culture results. There is a relatively small difference in how PCR and culture detected *C. albicans*, for example in 2022, 1.7% and 0.7% with PCR and culture, respectively. Additionally, the sample should always be taken from an anatomical site that appears clinically infected. According to

**TABLE 2** The mean TATs of culture and PCR methods from sample request to results from 2019 to 2022[a]

| Year | Culture | | PCR | |
|---|---|---|---|---|
| | h | D | h | D |
| 2019 | 603 | 25 | 16 | 0.7 |
| 2020 | 415 | 17 | 17 | 0.7 |
| 2021 | 420 | 17 | 16 | 0.7 |
| 2022 | 437 | 18 | 16 | 0.7 |
| Average | 469 | 19 | 16 | 0.7 |

[a]h = hours, D = days.

Hayette et al., a colonization of *C. albicans* on the skin close to the nails is unusual, and *C. albicans* result from nails is generally responsible for infection (19).

In conclusion, the number of nucleic acid-based detection samples has increased rapidly replacing traditional methods for analyzing superficial infection samples in Northern Finland. When used in the correct context DG covers well the most important pathogens and gives fast and accurate results. In addition, DG significantly decreased TAT from sample request to results. However, it is important to acknowledge its limitations. Most importantly, it currently does not replace traditional methods in identification of less common fungal pathogens. Although PCR-based method reduces the working time as well as unnecessary antifungal treatments, the exact overall cost-effect of PCR-based diagnostics of dermatophytes on healthcare system requires further studies. Future endeavors specifying whether particular specimen types or patient groups are associated with certain dermatophytes will be needed.

## ACKNOWLEDGMENTS

I. S. Junttila is funded by Tampere Tuberculosis Foundation, the Competitive State Research Financing of the Expert Responsibility Area of NordLab (grant KT0016) and Fimlab (grant X51409).

## AUTHOR AFFILIATIONS

[1]Northern Finland Laboratory Center, Nordlab, Oulu, Finland

[2]Department of Clinical Microbiology, HUS Diagnostic Center, University of Helsinki and Helsinki University Hospital, Helsinki, Finland

[3]Department of Dermatology and Medical Research Center Oulu, PEDEGO Research Unit, University of Oulu, Oulu University Hospital, Oulu, Finland

[4]Research Unit of Biomedicine and Internal Medicine, University of Oulu, Oulu, Finland

[5]Cytokine Biology Research Group, Tampere University, Tampere, Finland

[6]Fimlab Laboratories, Tampere, Finland

## AUTHOR ORCIDs

E. Aho-Laukkanen  http://orcid.org/0009-0006-2138-7242

## FUNDING

| Funder | Grant(s) | Author(s) |
| --- | --- | --- |
| Tampereen Tuberkuloosisäätiö (Tampere Tuberculosis Foundation) | | Junttila I. S. |
| The Competitive State Research Financing of the Expert Responsibility Area of NordLab | KT0016 | Junttila I. S. |
| Fimlab | 51409 | Junttila I. S. |

## AUTHOR CONTRIBUTIONS

E. Aho-Laukkanen, Investigation, Methodology, Project administration, Writing – original draft, Writing – review and editing | V. Mäki-Koivisto, Methodology, Project administration, Supervision, Visualization, Writing – original draft, Writing – review and editing | J. Torvikoski, Writing – original draft, Writing – review and editing | S. P. Sinikumpu, Writing – original draft, Writing – review and editing | L. Huilaja, Project administration, Supervision, Validation, Visualization, Writing – original draft, Writing – review and editing | I. S. Junttila, Validation, Visualization, Writing – original draft, Writing – review and editing, Investigation, Methodology, Project administration, Supervision

## ADDITIONAL FILES

The following material is available online.

## Supplemental Material

**Table S1 (Spectrum01049-24-s0001.docx).** DermaGenius PCR validation data.

## Open Peer Review

**PEER REVIEW HISTORY (review-history.pdf).** An accounting of the reviewer comments and feedback.

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
