## [Reviewer comments · Microbiology Spectrum]

Microbiology Spectrum

PCR enables rapid detection of dermatophytes in practice

Elina Aho-Laukkanen, Vesa Mäki-Koivisto, Jukka Torvikoski, Suvi-Päivikki Sinikumpu, Laura Huilaja, and Ilkka Junttila

Corresponding Author(s): Elina Aho-Laukkanen, Northern Finland Laboratory Center (NordLab)

Review Timeline:

Submission Date:	April 26, 2024
Editorial Decision:	May 23, 2024
Revision Received:	June 26, 2024
Editorial Decision:	July 5, 2024
Revision Received:	July 26, 2024
Accepted:	August 15, 2024

Editor: Vera Tesic

Reviewer(s): The reviewers have opted to remain anonymous.

Transaction Report:

DOI: <https://doi.org/10.1128/spectrum.01049-24>

Re: Spectrum01049-24 (PCR enables rapid detection of dermatophytes in practice)

Dear Mrs. Elina Aho-Laukkanen:

Thank you for the privilege of reviewing your work. Below you will find my comments, instructions from the Spectrum editorial office, and the reviewer comments.

Revision Guidelines

Sincerely,
Vera Tesic
Editor
Microbiology Spectrum

Reviewer #1 (Comments for the Author):

Thank you for the opportunity to review this article, which describes the implementation of a commercial dermatophyte PCR assay

Major comments:

Specimens have been examined either by Dermatophyte PCR OR microscopy and culture. I can't see that there has been an

analysis of samples being tested by all tests in parallel. How can we be certain of the accuracy of identifications made by the PCR? Or if performed in a previous study, this has not been cited.

Further, it is my experience that direct microscopy remains indispensable when PCR is performed, for detecting non-dermatophyte pathogens e.g. *Candida* and *Malassezia* yeasts, scabies mites, and non-dermatophyte onychomycoses. Not only that but how relevant is a positive *C. albicans* PCR result from skin and nails when there is no microscopy to assess whether this represents clinically significant overgrowth, or merely skin flora? I recommend providing additional analyses if possible, and discussion of this issue.

Line 180-188: the authors suggest that patients may be given antifungal therapy even in the absence of a positive culture or PCR result? Is this the case? It is interesting, because this is at odds with the advice of guidelines, at least for oral therapy. Further, while the authors point out that a positive PCR result while the patient has been taking antifungals can detect DNA from non-viable fungal cells, this is also a pitfall. How can positive PCR results differentiate active infection from a previous/cured infection, and what value does this result provide to the clinician? This is where microscopy results would be useful! The authors must discuss this issue further.

There is no comparison of the results in this study with those of other dermatophyte PCR studies, and this is necessary.

Minor comments:

Line 34: "timely verification of fungal infection is important, due to increasing global terbinafine resistance". There is a disconnect in this statement. Diagnosis of fungal infection has little bearing on the sensitivity or not to terbinafine - and resistance is not discussed in the manuscript. I suggest rewording this.

Lines 48-49, 65: There is no need to indicate (T.), (M.), (E.) or (C.) as an abbreviation for these genera. This is sufficiently understood globally to not require explanation.

Line 48: when describing the dermatophyte genera causing infection, the authors need to mention the new taxonomic groups of dermatophytes, including *Arthroderma* (e.g. *Arthroderma benhamiae*, referred to by its old name as *Trichophyton benhamiae* in this manuscript), and *Nannizzia*, *Paraphyton*, *Lophophyton* (these three being former *Microsporum* species). (de Hoog et al., 2017 doi: 10.1007/s11046-016-0073-9).

Lines 88-93: Does this extraction method follow the instructions/ recommendations of the DermaGenius manufacturer? If so, please state this. If not, indicate what the differences are.

Line 107: Does the direct microscopy procedure involve digestion of the skin/nail tissue with potassium hydroxide (it should). Please state this.

Line 109: If sample was too scant for both culture and microscopic examination, most guidelines would recommend prioritising the microscopy as this is more sensitive than culture (when stained with KOH-calcofluor white). Could the authors explain why culture was prioritised over microscopy?

Table 1: Please correct the spelling of 1) "*Trichophyton*". What was the identification of the "other dermatophytes" detected by culture?

Table 2: what is meant by h and D? Please include abbreviations. Can the authors provide an explanation as to the notable decrease in TAT of cultures after 2019? Is this related to having fewer cultures to manage, as more were examined by PCR instead?

Reviewer #2 (Comments for the Author):

Comments for the Author:

In this manuscript, authors describe the transition from a culture-based method to PCR to diagnose dermatophyte infections. The turnaround time for the PCR-based method was shorter compared to culture, and the positivity rate of the PCR-based method was higher than culture. I hope the authors view these suggestions as being constructive and designed to help strengthen their manuscript.

Major Comments:

Lines 149-150: Did the decreased turnaround time observed for PCR have any impact on clinical care (e.g., alterations in antifungal therapy, changes in clinical management)?

Lines 168-171: What accounts for the shift in the preferred method for detecting dermatophytes? Did the clinical guidance change for the diagnosis of dermatophyte infections or did the lab recommend PCR preferentially to culture? Was introduction of the DermaGenius assay based on clinician requests?

Lines 173-174: While it is very likely that PCR detected more organisms, was there any investigation of possible contamination

or false positive detections, which would also result in increased PCR detection? It would be helpful to include an analysis of how many culture and PCR results for the same specimen were concordant and how many were discordant in the results and discussion sections. The discussion might include how discordant results were resolved and how discordant results were interpreted by the clinician. Were there increased detections of dermatophytes associated with a specific specimen type (hair, skin, or nails) or in a specific patient population (immunocompromised vs immunocompetent, previous antifungal exposure vs none)? What are the performance characteristics of the DermaGenius assay (e.g., sensitivity, specificity, accuracy)?

Lines 187-188: Recommend describing the specific metrics where the clinical usefulness of DermaGenius has been evaluated.

Lines 190-199: Recommend including in the discussion on what the clinical significance is of the other organisms identified by culture alone.

Minor Comments:

Line 22: Recommend replacing "catchment area of approximately 720,000 people" with the number of individuals/samples that were included for testing in this study.

Lines 48, 49, and 65: Not necessary to include abbreviation for genera in name.

Line 147: It is not clear whether this sentence is meant to highlight the most common species identified by culture, PCR, or culture and PCR.

Responses to Reviewers

Reviewer #1 (Comments for the Author):

Thank you for the opportunity to review this article, which describes the implementation of a commercial dermatophyte PCR assay

Author: We thank Reviewer #1 for insightful and constructive comments on our work, we believe addressing them has improved the MS.

Major comments:

Specimens have been examined either by Dermatophyte PCR OR microscopy and culture. I can't see that there has been an analysis of samples being tested by all tests in parallel. How can we be certain of the accuracy of identifications made by the PCR? Or if performed in a previous study, this has not been cited.

ANSWER: This is a very important point raised by the reviewer. Basically, we did not exactly compare the methods, but rather, the different ways how clinical laboratory may support clinicians in laboratory diagnostics of dermatophyte infections. Initially, our transition from culture and microscopy to PCR in dermatophyte infection diagnostics was based on the clinical requirement for quicker responses that was provided by PCR. Before we introduced PCR to clinical diagnostics, the performance of PCR was verified by using fungal ATCC strains, fungal quality control strains and patient samples which were initially examined with culture and microscopy. Also, the DermaGenius PCR test is a CE-IVD assay which is validated by the manufacturer. In addition, the specificity of the positive result in PCR analysis is always verified by melting curve analysis of the PCR reaction. A short description on how accuracy of the PCR was evaluated is added to the revised manuscript on lines 105-107.

Further, it is my experience that direct microscopy remains indispensable when PCR is performed, for detecting non dermatophyte pathogens e.g. *Candida* and *Malassezia* yeasts, scabies mites, and non-dermatophyte onychomycoses. Not only that but how relevant is a positive *C. albicans* PCR result from skin and nails when there is no microscopy to assess whether this represents clinically significant overgrowth, or merely skin flora? I recommend providing additional analyses if possible, and discussion of this issue.

ANSWER: This again is an important and valid point raised by the reviewer. It is important that the sample is taken from an anatomical site that appears clinically infected. When looking at our data, there is a relatively small difference on how PCR and culture identify *C. albicans*, for example in 2022 1.7% with PCR and 0.7% with culture. In quite striking contrast for *T. rubrum* in similar comparison is 34.4% with PCR vs 2.2 with culture. We have included a discussion on this issue in the Discussion part of the revised MS on lines 208-210 and 217-223.

For the detection of non-dermatophyte pathogens, we use traditional methods. Additionally, we recommend that if PCR test gives a negative result and clinical suspicion of fungal infection is strong, the culture sample must be taken to detect rare pathogens. This is now mentioned on lines 211-213 of the revised MS.

Line 180-188: the authors suggest that patients may be given antifungal therapy even in the absence of a positive culture or PCR result? Is this the case? It is interesting, because this is at odds with the advice of guidelines, at least for oral therapy. Further, while the authors point out that a positive PCR result while the

patient has been taking antifungals can detect DNA from non-viable fungal cells, this is also a pitfall. How can positive PCR results differentiate active infection from a previous/cured infection, and what value does this result provide to the clinician? This is where microscopy results would be useful! The authors must discuss this issue further.

ANSWER: This again is a very adequate comment. Our wording in the initial manuscript was somewhat loose and we do apologize for this. National Finnish guideline does recommend topical treatment of tinea infections between toes without diagnostic sampling. For all other instances, culture or PCR sample is recommended to identify the causative agent. The rapid result obtained with PCR would decrease clinical use of those empirical antifungal treatments that are initiated while the clinicians are waiting for the culture results. We have now rephrased this in the revised MS (lines 187-191). Furthermore, the point of DNA from non-viable fungal cells is valid and underlines the fact that PCR is not suitable for following the efficacy of antifungal treatment. Discussion on this issue has been added to lines 194-196 in the revised MS.

There is no comparison of the results in this study with those of other dermatophyte PCR studies, and this is necessary.

ANSWER: This is an important point raised by the reviewer. We have now compared our results with other dermatophyte PCR studies on lines 198-203 and 210-211.

Minor comments:

Line 34: "timely verification of fungal infection is important, due to increasing global terbinafine resistance". There is a disconnect in this statement. Diagnosis of fungal infection has little bearing on the sensitivity or not to terbinafine - and resistance is not discussed in the manuscript. I suggest rewording this.

ANSWER: We have removed this sentence in the revised MS. However, based on our experience, *T. interdigitale* result with DermaGenius PCR test from elsewhere than the foot is actually *T. indotinae* species with culture and sequencing, thus indicating terbinafine resistance.

Lines 48-49, 65: There is no need to indicate (T.), (M.), (E.) or (C.) as an abbreviation for these genera. This is sufficiently understood globally to not require explanation.

ANSWER: We have removed the abbreviations in the revised MS, this issue was also raised by the reviewer #2.

Line 48: when describing the dermatophyte genera causing infection, the authors need to mention the new taxonomic groups of dermatophytes, including Arthroderma (e.g. Arthroderma benhamiae, referred to by its old name as Trichophyton benhamiae in this manuscript), and Nannizzia, Paraphyton, Lophophyton (these three being former Microsporum species). (de Hoog et al., 2017 doi: 10.1007/s11046-016-0073-9).

ANSWER: We have revised the MS accordingly on lines 49-50 of the revised MS.

Lines 88-93: Does this extraction method follow the instructions/ recommendations of the DermaGenius manufacturer? If so, please state this. If not, indicate what the differences are.

ANSWER: This is an important question from the reviewer #1. As mentioned in IFU of the DermaGenius, the extraction methods verified by the customer can be used for this method. We have been using the QIASymphony DSP Virus/Pathogen Mini kit for extraction since 2015 when in-house PCR test was validated with the manufacturer. The same extraction method was verified for DermaGenius PCR assay with the manufacturer. We have added the sentence of this topic on lines 94-95 in the revised MS.

Line 107: Does the direct microscopy procedure involve digestion of the skin/nail tissue with potassium hydroxide (it should). Please state this.

ANSWER: The direct microscopy method used does include digestion of skin/nail with potassium hydroxide. Statement indicating this has been added to line 113 of the revised MS.

Line 109: If sample was too scant for both culture and microscopic examination, most guidelines would recommend prioritising the microscopy as this is more sensitive than culture (when stained with KOH-calcofluor white). Could the authors explain why culture was prioritised over microscopy?

ANSWER: This is an important point raised by the reviewer. To our understanding, prioritizing culture or microscopy can vary depending on laboratory/country. In Finland, laboratories prioritize culture over microscopy according to Finnish National guidance Committee on Dermatophyte care and diagnostics (reference 8 in the revised MS). Fungal structures/mycelium in microscopic examination might also indicate a contamination which would result in false positive interpretation of microscopic evaluation of the sample. It is also good to clarify that, in practice, samples that have not been examined with microscope have been skin scrap samples, where the sample material is so finely divided and scarce that it has almost been impossible to see the sample material with naked eye. In these situations, the sample jar has been smeared with a moistened cotton swab to obtain a sample from the jar, and after this, the sample from swab has been cultured on agar plates. In all other cases, samples are divided into both microscopic examination and culture.

Table 1: Please correct the spelling of 1) "Thrichophyton". What was the identification of the "other dermatophytes" detected by culture?

ANSWER: Spelling has been fixed. Other dermatophytes included dermatophyte without further identification, and *Microsporum* sp. These have been added separately and the amounts of them have been corrected to Table 1 in the revised MS.

Table 2: what is meant by h and D? Please include abbreviations. Can the authors provide an explanation as to the notable decrease in TAT of cultures after 2019? Is this related to having fewer cultures to manage, as more were examined by PCR instead?

ANSWER: h and D stand for hours and days. Abbreviations have been added to Table 2 in the revised MS. As for the notable decrease in culture TATs after 2019, this is a very interesting point. It is not likely that the culture management was easier because of fewer samples. Rather, outsourcing of culture samples from NordLab to Huslab occurred in 5/2019 and it is most likely, that the culture process in Huslab was somewhat faster and explains the observed difference.

Reviewer #2 (Comments for the Author):

Comments for the Author:

In this manuscript, authors describe the transition from a culture-based method to PCR to diagnose dermatophyte infections. The turnaround time for the PCR-based method was shorter compared to culture, and the positivity rate of the PCR-based method was higher than culture. I hope the authors view these suggestions as being constructive and designed to help strengthen their manuscript.

Author: We thank reviewer #2 for valuable, constructive, and insightful comments. We believe addressing them has substantially improved and clarified the MS.

Major Comments:

Lines 149-150: Did the decreased turnaround time observed for PCR have any impact on clinical care (e.g., alterations in antifungal therapy, changes in clinical management)?

ANSWER: This is a very important question the reviewer#2 raises; what is the impact of the switch of the diagnostic tool to actual patient care? We believe that the one important benefit of PCR on clinical care is most visible in cases where antifungal treatment is initiated without result of the relatively slow culture method. This is also important when considering the loose use of antifungals while global resistance to terbinafine is increasing. See also our answer #3 to reviewer #1.

Lines 168-171: What accounts for the shift in the preferred method for detecting dermatophytes? Did the clinical guidance change for the diagnosis of dermatophyte infections or did the lab recommend PCR preferentially to culture? Was introduction of the DermaGenius assay based on clinician requests?

ANSWER: This again important point by the reviewer, why the change of the diagnostic tool was made in the first place? The shift for PCR in our area was based on the clinicians' need for more rapid answers in identifying dermatophytes. It is of interest that since 2021, the Finnish National guidance Committee on Dermatophyte care and diagnostics has included PCR as a diagnostic tool for these infections.

Lines 173-174: While it is very likely that PCR detected more organisms, was there any investigation of possible contamination or false positive detections, which would also result in increased PCR detection? It would be helpful to include an analysis of how many culture and PCR results for the same specimen were concordant and how many were discordant in the results and discussion sections. The discussion might include how discordant results were resolved and how discordant results were interpreted by the clinician. Were there increased detections of dermatophytes associated with a specific specimen type (hair, skin, or nails) or in a specific patient population (immunocompromised vs immunocompetent, previous antifungal exposure vs none)? What are the performance characteristics of the DermaGenius assay (e.g., sensitivity, specificity, accuracy)?

ANSWER: This a very critical point concerning this MS, how can one trust the PCR results? As pointed out in our response to point 1 by Reviewer #1 we introduced this method upon clinical verification with ATCC specimens, fungal quality control strains, and patient samples, based on our ISO 15189 Certified laboratory practices. In addition to this, there are two things that we believe further improve the specificity of PCR in clinical diagnostics. These are 1) the sample should always be taken from a site that appears clinically infected and 2) dermatophytes are not part of normal microbiome of skin, nail, or mucosal surfaces. In this study, we did not examine the same sample parallel with PCR and culture, expect we compared the overall results of both methods. As pointed out by the reviewer, it would be very interesting to examine more

specifically, if there are increased detections of dermatophytes associated with a specific specimen type or in a specific patient population. This is added on lines 230-231 in revised MS.

There are no exact values of sensitivity, specificity and accuracy described in the IFU. According to IFU of Dermagenius, the LoD or analytical sensitivity was determined for all the targets using a dilution series of genomic DNA. The final LoD was confirmed by testing 20 replicates with a positivity rate of $\geq 95\%$. The LoDs varied between 1.0-9.0 copies/ μl per target. The analytical specificity of the DermaGenius was determined by testing DNA of various clinically relevant nail, hair and skin pathogens including other dermatophyte species, various fungal strains, and bacteria.

For example, in Trovato et al. studies (2022 and 2023) the sensitivity, specificity, positive predictive value, and negative predictive value of DermaGenius in the cutaneous dermatophytosis were, respectively, 94.7%, 78.8%, 88.5%, and 89.6%, in addition to 78.5%, 100%, 100%, and 75.9%, respectively, for onychomycosis diagnosis.

Lines 187-188: Recommend describing the specific metrics where the clinical usefulness of DermaGenius has been evaluated.

ANSWER: Firstly, we apologize that we may not have correctly understood the meaning of "specific metrics", but based on our study, the number of PCR samples has been constantly increasing, which indicates that DermaGenius is clinically useful giving fast and accurate results.

Lines 190-199: Recommend including in the discussion on what the clinical significance is of the other organisms identified by culture alone.

ANSWER: This is an important point. The clinical significance of isolated cases of findings solely made by culture method (*Aspergillus* sp, *Scopulariopsis* sp, other moulds, *Fusarium* sp, dermatophyte, other *Microsporum* sp, *Acremonium* sp, other *Trichophyton* sp) is complex. While it is clear, that negative PCR result on patients with these microbes may delay the care of a single patient it is important to acknowledge that our instructions to clinicians specifically underline the importance of culture if PCR is negative and clinical condition strongly hint to fungal infection. We have discussed this issue on lines 211-217 in revised MS.

Minor Comments:

Line 22: Recommend replacing "catchment area of approximately 720,000 people" with the number of individuals/samples that were included for testing in this study.

ANSWER: We would still like to include the catchment area to give the reader a general impression of how common these infections are in our area. We have rephrased this and added the information of the ethnicity in the revised MS on lines 22-23. A total amount of samples has also been added on line 81 and table 1.

Lines 48, 49, and 65: Not necessary to include abbreviation for genera in name.

ANSWER: These have been removed.

Line 147: It is not clear whether this sentence is meant to highlight the most common species identified by culture, PCR, or culture and PCR.

ANSWER: This sentence has been rephrased on line 155 to indicate that overall (with PCR and culture) these are the most common dermatophytes identified.

Re: Spectrum01049-24R1 (PCR enables rapid detection of dermatophytes in practice)

Dear Mrs. Elina Aho-Laukkanen:

Thank you for the privilege of reviewing your work. Below you will find my comments, instructions from the Spectrum editorial office, and the reviewer comments.

Revision Guidelines

Sincerely,
Vera Tesic
Editor
Microbiology Spectrum

Reviewer #1 (Comments for the Author):

While the authors have responded to both reviewer's comments well, more could be done to incorporate these comments into the manuscript. The manuscript still seems to be an unbalanced discussion in favour of using one particular PCR assay over the traditional (gold standard) method of microscopy and culture, with limited discussion of the limitations of PCR.

1) Rather than just mentioning it to the reviewers, please include the validation and comparative data that was initially performed

to assess the performance of the PCR directly with microscopy and culture. Also discuss any other publications that have performed such studies using the DermaGenius assay. The fact that the assay is CE-IVD marked does not necessarily mean that it works perfectly in every laboratory.

2) Please discuss more explicitly the limitations of using PCR alone to assess specimens for pathogens, i.e. without microscopy. That is - that those pathogens that the assay is designed to detect can be detected, and *Malassezia*, scabies and non-dermatophyte onychomycoses will be missed. This places additional burden on both the patient and the clinician to follow up for further testing.

Reviewer #2 (Comments for the Author):

In this manuscript, authors describe the transition from a culture-based method to PCR to diagnose dermatophyte infections. This manuscript has been strengthened by clarifications to the text and additions to the discussion. I hope the authors view these minor suggestions as being constructive.

Comments:

Lines 105-107: Recommend adding any information on whether the verification, or experience since assay implementation, found any differences from what was expected in the performance of the DG assay. For example, in the response to reviewer #1 comments, the authors state, "However, based on our experience, *T. interdigitale* result with DermaGenius PCR test from elsewhere than the foot is actually *T. indotinae* species with culture and sequencing, thus indicating terbinafine resistance." It is my opinion that this information regarding DG PCR performance is very important to include in the manuscript. The authors have valuable experience with this assay that would benefit others as there is a clear consequence of misidentification of *T. interdigitale* with *T. indotinae* (associated terbinafine resistance).

Lines 198-203: Recommend adding the data from the Trovato et al. studies (2022 and 2023) on the sensitivity, specificity, positive predictive value, and negative predictive value of DermaGenius in cutaneous dermatophytosis as it relates to clinical usefulness.

Reviewer #1 (Comments for the Author):

While the authors have responded to both reviewer's comments well, more could be done to incorporate these comments into the manuscript. The manuscript still seems to be an unbalanced discussion in favour of using one particular PCR assay over the traditional (gold standard) method of microscopy and culture, with limited discussion of the limitations of PCR.

RESPONSE: We thank the reviewer for positive comments on our response to initial critique on our MS. We would like to stress here, that we do not favor PCR over culture in diagnostics of dermatophyte infections. Rather, our focus in this work is to describe how clinician-initiated introduction of PCR has changed the number of cultures and PCR tests over the years in our area. We focus on one commercial test simple because this test was selected for clinical use in our laboratory. We reiterate, that we have no commercial interests or other conflicts of interest.

1) Rather than just mentioning it to the reviewers, please include the validation and comparative data that was initially performed to assess the performance of the PCR directly with microscopy and culture. Also discuss any other publications that have performed such studies using the DermaGenius assay. The fact that the assay is CE-IVD marked does not necessarily mean that it works perfectly in every laboratory.

RESPONSE: This is a valid point. It is evident, that CE-IVD marking does not guarantee that an assay would work consistently everywhere. Validation data from our laboratory is now provided in Supplementary Table 1. Works that have used DermaGenius are already cited in the MS (citations 13-15 and 19), but discussion on the performance is added to lines 177-181 and 207-211 of the Discussion of the revised MS. See also our response to reviewer #2.

2) Please discuss more explicitly the limitations of using PCR alone to assess specimens for pathogens, i.e. without microscopy. That is - that those pathogens that the assay is designed to detect can be detected, and *Malassezia*, scabies and non-dermatophyte onychomycoses will be missed. This places additional burden on both the patient and the clinician to follow up for further testing.

RESPONSE: To address this, we rephrased the paragraph on line 221 and added a discussion of limitations to lines 234-237. Finnish guidelines concerning *Malassezia* (yeast) and scabies (*Sarcoptes scabiei*, a mite) currently favor clinical diagnosis, without laboratory verification of the causative microbe. However, if *Malassezia* is found on culture sample, this information is conveyed to clinics, but as said, *Malassezia* should not be actively sought by culture in diagnostic sense, according to Finnish guidelines.

Reviewer #2 (Comments for the Author):

In this manuscript, authors describe the transition from a culture-based method to PCR to diagnose dermatophyte infections. This manuscript has been strengthened by clarifications to the text and additions to the discussion. I hope the authors view these minor suggestions as being constructive.

RESPONSE: We thank the reviewer for constructive comments on our work.

Comments:

Lines 105-107: Recommend adding any information on whether the verification, or experience since assay implementation, found any differences from what was expected in the performance of the DG assay. For example, in the response to reviewer #1 comments, the authors state, "However, based on our experience, *T. interdigitale* result with DermaGenius PCR test from elsewhere than the foot is actually *T. indotinae* species with culture and sequencing, thus indicating terbinafine resistance." It is my opinion that this information regarding DG PCR performance is very important to include in the manuscript. The authors have valuable experience with this assay that would benefit others as there is a clear consequence of misidentification of *T. interdigitale* with *T. indotinae* (associated terbinafine resistance).

RESPONSE: We have now added a discussion on our validation data and Supplementary Table 1. For the time being, our observation of *T. interdigitale*/*T. indotinae* misidentification based on anatomical location is purely observational and we chose to omit discussing this in detail. However, we are in process of following this aspect more consistently in a future work. As the reviewer points out, this is an important aspect.

Lines 198-203: Recommend adding the data from the Trovato et al. studies (2022 and 2023) on the sensitivity, specificity, positive predictive value, and negative predictive value of DermaGenius in cutaneous dermatophytosis as it relates to clinical usefulness.

RESPONSE: We thank the reviewer for clearing this point, the data from Trovato et al. studies 2022 and 2023 have been added (lines 207-211 of the revised MS).

Re: Spectrum01049-24R2 (PCR enables rapid detection of dermatophytes in practice)

Dear Mrs. Elina Aho-Laukkanen:

Your manuscript has been accepted, and I am forwarding it to the ASM production staff for publication. Your paper will first be checked to make sure all elements meet the technical requirements. ASM staff will contact you if anything needs to be revised before copyediting and production can begin. Otherwise, you will be notified when your proofs are ready to be viewed.

Sincerely,
Vera Tesic
Editor
Microbiology Spectrum

Reviewer #2 (Comments for the Author):

No further comments. All questions have been addressed.